# Effects of perinatal mobile apps for couples on psychosocial and parenting outcomes: A systematic review and meta-analysis

Luis Ttito-Paricahua[1☉], Marlene Magallanes-Corimanya[2☉], Liz Mendoza-Aucaruri[1☉], Jean Pierre López-Mesia[3,4☉], Evelyn M. Asencios-Falcón[2‡], Alicia Lopez-Gomero[5‡], Alvaro Taype-Rondan[6,7]*

**1** Facultad de Medicina Humana, Universidad Científica del Sur, Lima, Peru, **2** Facultad de Obstetricia y Enfermería, Universidad de San Martín de Porres, Lima, Peru, **3** Facultad de Medicina Humana, Universidad Nacional de la Amazonía Peruana, Iquitos, Peru, **4** Hospital Regional de Loreto, Loreto, Peru, **5** Departamento de investigación, Universidad Continental, Lima, Peru, **6** EviSalud – Evidencias en Salud, Lima, Peru, **7** Unidad de Investigación para la Generación y Síntesis de Evidencias en Salud, Vicerrectorado de Investigación, Universidad San Ignacio de Loyola, Lima, Peru

☉ These authors contributed equally to this work.
‡ These authors also contributed equally to this work.
* alvaro.taype.r@gmail.com

## Abstract

The transition to parenthood involves significant changes, and while mobile apps offer promising perinatal support, their impact on couples' psychosocial and parenting outcomes remains uncertain. The objective of this systematic review was to assess the effects of perinatal mobile applications designed for couples on psychosocial well-being and parenting-related outcomes. To evaluate the impact of mobile apps on psychosocial and parenting outcomes during the perinatal period, a systematic review of randomized controlled trials (RCTs) was conducted. Our study protocol was registered in PROSPERO under the identifier CRD42024578397. Searches in major databases continued through November 2024. Two reviewers independently handled data extraction and bias assessment. Meta-analyses used random-effects models, and evidence certainty was evaluated using the GRADE approach. Four RCTs (n = 3592 parents) were included. At one month postpartum, perinatal mobile applications may have little to no effect on postnatal depression (SMD: –0.00; 95% CI: –0.17 to 0.16; low certainty), state anxiety (MD: –1.50; 95% CI: –3.93 to 0.93; low certainty), and parent-to-infant bonding (MD: –0.25; 95% CI: –0.92 to 0.42; low certainty). Similarly, little to no effect was found for breastfeeding self-efficacy (MD: 0.90; 95% CI: –2.30 to 4.10; low certainty) and partner support during breastfeeding (MD: 1.10; 95% CI: –2.48 to 4.68; low certainty). The evidence was very uncertain regarding their effects on perceived parenting self-efficacy, parenting satisfaction and social support. These findings indicate that perinatal mobile applications may have limited impact on psychosocial and parenting outcomes in the early postpartum period. Further high-quality studies with longer follow-up are needed to clarify their effectiveness.

**Data availability statement:** All data underlying the findings of this study are fully available within the Supporting Information materials. Specifically, the extracted data used in the systematic review and meta-analysis are provided in S3 Table. Data extraction sheet for included studies. The study protocol has been registered in PROSPERO (ID: CRD42024578397; https://www.crd.york.ac.uk/PROSPERO/) and is also available as Supporting Information material (S1 File).

**Funding:** The authors received no specific funding for this work.

**Competing interests:** The authors have declared that no competing interests exist.

## Introduction

The transition to parenthood is a critical period marked by psychological, emotional, and social changes for both mothers and fathers [1–3]. While maternal well-being during the perinatal period has been widely studied, increasing evidence highlights the importance of paternal mental health and involvement in caregiving, as it significantly influences maternal health, infant development, and overall family dynamics [4].

Parental stress, anxiety, and depressive symptoms are common during this period, affecting both parents' ability to bond with their infant and adapt to new family roles. Research suggests that early psychosocial interventions can mitigate these challenges, improving emotional well-being and parenting outcomes [5]. Digital health technologies, including mobile applications, have emerged as promising tools for delivering perinatal support by providing easily accessible resources [6–8].

While numerous mobile applications target maternal health, fewer are specifically designed to support fathers or couples during the perinatal period. Given that fathers increasingly seek information and support through digital platforms [9,10], mobile applications could serve as a valuable tool to promote paternal engagement and well-being [11]. However, previous systematic reviews have primarily focused on the effects of these interventions on mothers [12] and those that have considered their impact on couples have neither conducted meta-analyses nor assessed the certainty of the evidence [5,13,14].

Mobile applications, by providing accessible and personalized content, could serve as a promising tool to address fathers' needs and encourage their active involvement during the perinatal period. These platforms enable access to relevant information without geographical or time constraints and provide immediate data delivery when needed most [8].

Thus, this systematic review aims to assess the effects of perinatal mobile applications designed for couples on psychosocial well-being and parenting-related outcomes.

## Methods

### Study design and registration

We conducted a systematic review following the PRISMA 2020 guidelines for transparent and complete reporting [15], as detailed in the PRISMA checklist provided in S1 Appendix. Our study protocol was registered in the International Prospective Register of Systematic Reviews (PROSPERO) under the identifier CRD42024578397, as detailed in S1 File (https://www.crd.york.ac.uk/PROSPERO/).

### Research question and eligibility criteria

Among couples and parents in the perinatal period, what is the effect of mobile application–based interventions, compared to standard care or no intervention, on psychosocial and parenting-related outcomes? To address this question, we included randomized controlled trials (RCTs) involving couples and parents during the perinatal period, excluding studies that focused exclusively on mothers.

The intervention of interest was mobile applications (apps) designed to improve psychosocial or parenting-related outcomes, implemented during the perinatal period (from 22 weeks of pregnancy to 4 weeks postpartum). Eligible apps had to serve as the primary intervention, although secondary virtual or in-person components were permitted. Studies in which apps were used exclusively for communication (e.g., chat, video calls, or consultations with healthcare professionals) or for sending text messages were excluded.

The control group comprised participants who did not receive mobile app-based interventions, including those receiving standard care, no intervention, or other non-app-based support.

The review focused on psychosocial and parenting-related outcomes, such as postnatal depression, perceived self-efficacy, parenting satisfaction, social support, state anxiety, parent-to-infant bonding, breastfeeding self-efficacy and partner support during breastfeeding. Studies reporting only unrelated biochemical outcomes were excluded.

## Information sources and search strategy

We systematically searched PubMed/MEDLINE, Embase, Cochrane Library (CENTRAL), and Scopus from their inception until November 21, 2024, with no language restrictions. The search strategy comprised four main conceptual blocks: (1) population-related terms (e.g., parents, fathers, couples, pregnant women); (2) intervention terms related to mobile health (e.g., mHealth, smartphone, app, telemedicine); (3) outcomes related to mental health and parenting (e.g., depression, anxiety, bonding, self-efficacy); and (4) study design filters for randomized controlled trials. Detailed search strategies for each database are provided in the S1 Table. Additionally, we searched ClinicalTrials.gov for relevant clinical trial records and screened the reference lists of included studies and relevant systematic reviews [5,13,14].

## Selection and data collection process

Search results from electronic databases were imported into Rayyan (Qatar Computing Research Institute) for initial reference management [16,17]. After duplicate removal, six review authors (LMA, JPLM, LTP, MMC, EMAF, and ALG) worked in pairs to screen studies, first by evaluating titles and abstracts, followed by full-text review. Any discrepancies were resolved through discussion between the reviewers or, if needed, by consulting a third author (ATR).

Data from included studies were extracted using a standardized data extraction form. Six authors (LMA, JPLM, LTP, MMC, EMAF, and ALG) worked in pairs to independently extract study data, with disagreements resolved through discussion or, if necessary, with the mediation of a third author (ATR). When critical information was unclear, we contacted the original study authors for clarification. The complete data extraction table is provided in the supplementary file S3 Table.

## Variables

Information was collected from each selected study, considering its design, population, intervention, comparison, psychosocial well-being and parenting-related outcomes, as well as study funding. For the intervention with perinatal mobile applications, data were collected following the Template for Intervention Description and Replication (TIDieR) checklist [18].

Data collection was conducted systematically, including both continuous and categorical variables. The follow-up period was set between 4 and 6 weeks postpartum, based on the duration of the intervention in each study. When additional information was required, corresponding authors were contacted via email.

## Risk of bias assessment

To evaluate the risk of bias, we employed the Cochrane Risk of Bias Tool (RoB 1.0), which includes eight domains: random sequence generation, allocation concealment, blinding of participants, personnel, and outcome assessment, incomplete outcome data, selective reporting, and other potential sources of bias [19]. Six authors (LMA, JPLM, LTP, MMC, EMAF, and ALG) independently and in pairs assessed the risk of bias in the selected studies. Any disagreements were addressed through discussion or by consulting a third author (ATR).

## Statistical analyses

We conducted meta-analyses to synthesize the results of studies examining similar outcomes. For continuous outcomes measured using different scales or analyzed in varying ways, we calculated standardized mean differences (SMD) to facilitate comparisons and meta-analysis. Meta-analyses were performed using random-effects models in RevMan [20] and heterogeneity across studies was assessed using the $I^2$ statistic. When key data such as means or standard deviations were available only in graphical form, they were extracted using version 4.3.2 of the R software from the Foundation for Statistical Computing [21,22]. The R script used for extracting data from published figures is provided in the S2 Appendix.

## Certainty of the evidence assessment

We used the Grading of Recommendations, Assessment, Development, and Evaluations (GRADE) framework to assess the certainty of evidence for all outcomes [23], applying two thresholds based on minimally important differences (MIDs). The MID for SMDs was set at 0.2, following Cohen's d [24,25]. For state anxiety, we identified a MID of 10 points on the State-Trait Anxiety Inventory (STAI) scale based on available literature [26].

For other outcomes reported as mean differences (MDs), where no established MID was found in the literature, we determined MID values through informal consensus among the authors, incorporating clinical judgment to ensure they reflect meaningful changes in each outcome measure. The following MIDs were used: 2 points for parent-to-infant bonding, 5 points for breastfeeding self-efficacy and 7 points for partner support during breastfeeding.

To assess the certainty of the evidence, we followed the GRADE Handbook guidelines [23] and evaluated five key domains: risk of bias (using the Cochrane risk of bias tool), imprecision (considering whether confidence intervals included one or two MIDs), indirectness (by comparing our PICO question with those of the included studies), inconsistency (assessing whether studies' point results were distributed on one or both sides of an MID), and publication bias (examined in meta-analyses with at least 10 studies). Additionally, we used the Summary of Findings (SoF) table and adhered to GRADE guidelines for interpreting our results [27].

## Results

During the systematic database search, 3347 records were initially identified. After removing 1856 duplicates, 1491 titles and abstracts were screened, of which 1472 were excluded. A total of 19 full-text articles were assessed for eligibility, and four randomized controlled trials (RCTs) were ultimately included in the review [28–31]. No additional RCTs were identified through complementary search methods. A detailed list of excluded full-text articles is provided in S2 Table, and the study selection process is shown in (Fig 1).

## Studies characteristics

The included studies were published between 2016 and 2023. The total sample size comprised 3592 randomized parents. The studies were conducted across different regions of the world: one in America (The United States) [28], two in Asia (Singapore) [29,31] and one study in Oceania (Australia) [30].

The intervention characteristics varied across the studies. The mobile app was provided during the perinatal stage, defined as the period from 22 weeks of gestation to the first 4 weeks of neonatal life (28 days). The timing of outcome measurement ranged from immediately after the intervention to 6 weeks postpartum.

Regarding the comparators, three studies used routine or standard care provided by each institution as the control group [28,29,31]. In a single study [30], in addition to routine care, three additional interventions were implemented: one based on a mobile application, another non-virtual, and a combination of both, resulting in a four-arm design. For this systematic review, we assessed the results from the routine care and mobile application arms (Table 1).

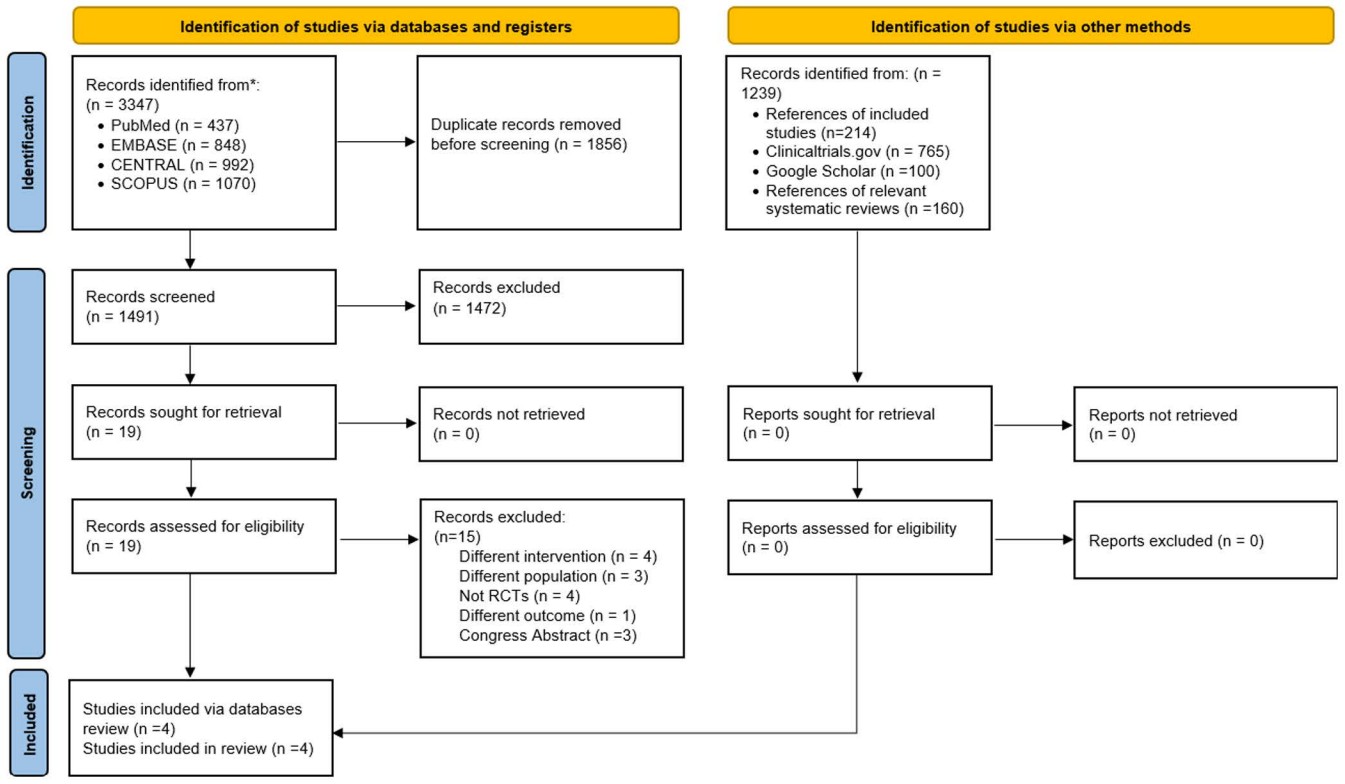

**Fig 1. Flow diagram of studies inclusion.**

Postnatal depression was assessed in two studies [29,31] using the Edinburgh Postnatal Depression Scale (EPDS), a 10-item instrument in which higher scores indicate greater severity of depressive symptoms [32]. Similarly, perceived self-efficacy was measured in the same two studies using the Parenting Efficacy Scale (PES), a 10-item questionnaire where higher scores reflect greater perceived parenting self-efficacy [33].

Parenting satisfaction was assessed in three studies. One study [28] used the Parenting Sense of Competence Scale (PSOC), a 17-item instrument where higher scores suggest greater parenting self-efficacy and satisfaction [34]. In contrast, two studies [29,31] employed the satisfaction subscale of the What Being the Parent of a New Baby is Like (WPBL) scale [35].

Social support during the perinatal period was assessed in two studies [29,31] using the Perceived Social Support for Parenting (PSSP) scale. This 8-item tool measures parents' satisfaction with the social support received from their partners and others, with higher scores indicating greater perceived support [36].

State anxiety was assessed using the State-Trait Anxiety Inventory-State subscale (STAI-Y1) [31], a 20-item instrument rated on a 4-point Likert scale, with total scores ranging from 20 to 80. Higher scores indicate greater levels of anxiety [37].

Parent-infant bonding was assessed using the Parent-to-Infant Bonding Questionnaire (PIBQ) [31], an 8-item tool rated on a 4-point Likert scale, where higher scores (range: 0–21) indicate poorer bonding [38].

Breastfeeding self-efficacy and postpartum partner support were assessed in one study [30] using the Breastfeeding Self-Efficacy Scale-Short Form (BSES-SF) [39] and the Postpartum Partner Support Scale (PPSS) [40], respectively. The BSES-SF includes 14 items evaluating maternal confidence in breastfeeding, while the PPSS (25-item version) measures emotional, informational, and instrumental support from the partner, both with higher scores indicating stronger outcomes.

**Table 1. Studies characteristics.**

| Author (Country, Year) | Population, Age, Gestational Age at recruitment, Education | Number of randomized participants | Intervention: Perinatal mobile apps | Intervention, usage | Control group | Funding |
|---|---|---|---|---|---|---|
| Garfield FC (United States, 2016) [28] | Parents<br>Age: 33.7±5.8 years<br>Gestational age: 29.7 weeks<br>Education: employed with complete university: 81.7% | 90 parents (I: 44, C: 46) | NICU-2-Home App: Developed to enhance parental self-efficacy during NICU hospitalization and transition to home. It includes: (1) Passport-2-Home, a discharge checklist; (2) Education Center, multimedia content on NICU infant care; (3) Baby Connect, an app for tracking daily care activities; and (4) Mood Tracker, a feature to monitor parental emotional well-being. | Intervention: Parents received a smartphone with the app, with free access for four weeks, covering the last two weeks in the NICU and the first two at home. Usage: Usage was highest at the beginning and declined over time. 26% were above-average (mean usage: 9.7 times/day), 48% average (3.8 times/day), and 26% below-average (1.3 times/day) users. | Standard NICU care, including printed handouts and educational guidance from nurses before discharge. | Agency for Health Research and Quality (R21 HS20316) |
| Shorey S (Singapore, 2017) [29] | Parents<br>Age: 33±5 years<br>Gestational age: 38.84±1.26 weeks<br>Education: 72.4% university | 125 couples (I: 63, C: 62) | Home-but not alone app: Developed to provide psychoeducational support through role-specific educational content, periodic push notifications with timely information, and asynchronous communication with healthcare professionals. Educational materials included PDFs on newborn and maternal care, instructional videos on tasks like baby bathing and breastfeeding, and audio files for mothers with reading restrictions during the confinement period. | Intervention: Parents were given access to the app after hospital discharge for postnatal psychoeducational support. Weekly reminders were sent to encourage usage, and data collection occurred at baseline and four weeks postpartum. The research team monitored usage and sent weekly reminders. Usage: varying frequency. | Routine postnatal care, including educational support from nurses and midwives during hospitalization and a scheduled medical appointment postpartum. | National University of Singapore Start-up Grant (NUHSRO/2015/063/SU/01) |
| Scott JA (Australia, 2021) [30] | Couples<br>Age: 33±5 years<br>Gestational age: NR<br>Education: 59% Some/completed university | 1426 couples (Ia:338, Ib: 397, Ic: 333, C: 358)[a] | Milk Man App: A mobile application designed to engage fathers in breastfeeding support through gamification, peer interaction, and educational content. The app features (1) a discussion forum for fathers to share experiences and advice, (2) twice-weekly push notifications linking to polls and conversation starters, and (3) a library with breastfeeding-related information. | Intervention: The app was available from 32 weeks of gestation to 6 months postpartum, featuring gamification, a forum, and twice-weekly notifications. Usage: A total of 79,000 interactions were recorded, with peak activity around birth. On average, each user read 11.5 articles, posted 2.21 comments, and voted in 6 polls. | Usual care, consisting of their participation in the breastfeeding component of hospital-based antenatal classes designed for couples. | Western Australia Health Promotion Foundation (Healthway grant 24023) |
| Shorey S (Singapore, 2023) [31] | Heterosexual married couples<br>Age: 31±5 years<br>Gestational age: NR<br>Education: 76% Some/completed university | 200 couples (I: 100, C: 100) | SPAApp: A mobile psychoeducational intervention designed to provide emotional and informational support to expectant and new parents. The app includes theoretical and evidence-based content, covering topics such as perinatal mental health, newborn care, and postpartum adjustment. Parents in the intervention group were also matched with peer volunteers who provided additional support. | Intervention: Parents received access to the app. Periodic notifications were sent, personalized according to the pregnancy stage or the baby's age, starting at 24 weeks of gestation and continuing until 6 months postpartum. Usage: varied among participants, with a 28.5% dropout rate over 12 months. | Standard perinatal care, which included antenatal check-ups, optional prenatal educational classes, in-hospital postpartum care, and a follow-up medical review at 6 weeks postpartum. | Ministry of Health (HSRGHP18may-0001) |

NR: Not reported; I: Intervention; C: Control; NICU: Neonatal Intensive Care Unit; SPA:Supportive Parenting App.

[a](Ia: Father-Focused Antenatal Breastfeeding Class (FFABC); Ib: Milk Man; Ic: Combination).

PLOS Mental Health

## Risk of bias

The most affected risk of bias domain was participant blinding (4/4), followed by blinding of personnel (3/4), blinding of outcome assessment (3/4), and incomplete outcome data (3/4). Additionally, Other bias was present in 2/4 studies, while Allocation concealment was affected in 1/4 studies. In contrast, Random sequence generation and Selective reporting were consistently assessed as low risk across all studies (Fig 2).

## Results of syntheses and certainty of the evidence

The Summary of Findings (SoF) table is presented in Table 2, and the meta-analysis results are shown in Figs 3–6.

Perinatal mobile applications may result in little to no difference in postnatal depression at 1 month (2 RCTs; SMD: –0.00; 95% CI: –0.17 to 0.16; low certainty), and may also result in little to no difference in state anxiety at 1 month (1 RCT; MD: –1.50; 95% CI: –3.93 to 0.93; low certainty). Similarly, they may result in little to no difference in parent-to-infant bonding at 1 month (1 RCT; MD: –0.25; 95% CI: –0.92 to 0.42; low certainty), and in breastfeeding self-efficacy at 4–6 weeks (1 RCT; MD: 0.90; 95% CI: –2.30 to 4.10; low certainty). Partner support during breastfeeding at 4–6 weeks also showed little to no difference when using perinatal mobile applications (1 RCT; MD: 1.10; 95% CI: –2.48 to 4.68; low certainty).

The evidence is very uncertain about the effect of perinatal mobile applications on perceived parenting self-efficacy at 1 month (2 RCTs; SMD: 0.56; 95% CI: –0.37 to 1.49; very low certainty), parenting satisfaction at 1 month (3 RCTs; SMD: 1.05; 95% CI: –0.43 to 2.52; very low certainty), and social support at 1 month (2 RCTs; SMD: 0.75; 95% CI: –0.16 to 1.65; very low certainty).

# Discussion

## Summary of main results

Perinatal mobile applications may have little to no effect on postnatal depression, state anxiety, parent-to-infant bonding, breastfeeding self-efficacy, and partner support during breastfeeding. Additionally, the evidence is very uncertain regarding

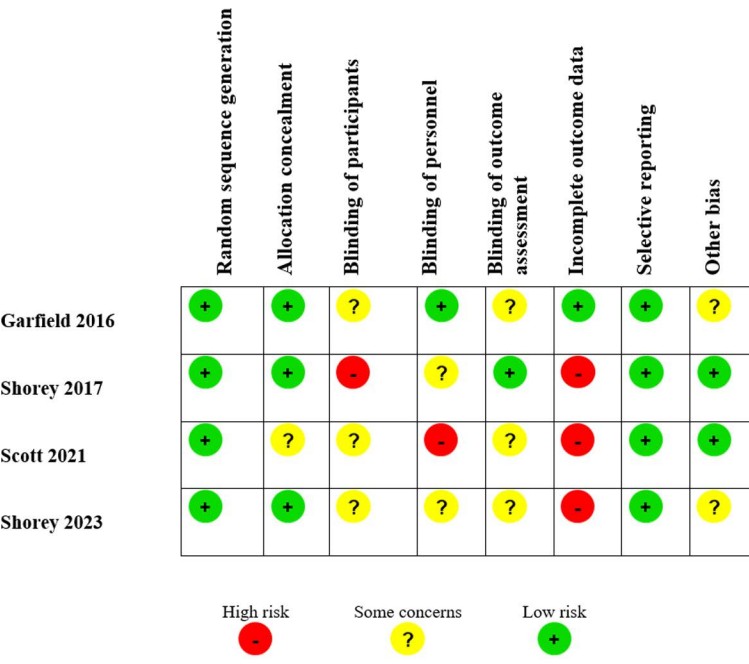

**Fig 2. Risk of bias.**

**Table 2. Summary of findings table for the effects of perinatal mobile applications (Apps) on the psychosocial and parenting outcomes of couples.**

| Outcomes (follow-up after delivery) | Number of patients (number of studies) | Perinatal mobile applications group | Control group | Absolute difference (95% CI) | Certainty | Interpretation |
|---|---|---|---|---|---|---|
| Postnatal Depression [29,31] (1 month) | 577 (2 RCTs) | n = 302 | n = 275 | SMD: -0.00 (-0.17, 0.16) | ⊕⊕◯◯ Low (a) | Perinatal mobile apps may result in little to no difference in postnatal depression at 1 month. |
| Perceived Self-Efficacy [29,31] (1 month) | 577 (2 RCTs) | n = 302 | n = 275 | SMD: 0.56 (-0.37, 1.49) | ⊕◯◯◯ Very low (a,b) | The evidence is very uncertain about the effect of perinatal mobile apps on perceived self-efficacy at 1 month. |
| Parenting satisfaction [28,29,31] (1 month) | 638 (3 RCTs) | n = 332 | n = 306 | SMD: 1.05 (-0.43, 2.52) | ⊕◯◯◯ Very low (a,b) | The evidence is very uncertain about the effect of Perinatal Mobile Apps on parenting satisfaction at 1 month. |
| Social support [29,31] (1 month) | 577 (2 RCTs) | n = 302 | n = 275 | SMD: 0.75 (-0.16, 1.65) | ⊕◯◯◯ Very low (a,c) | The evidence is very uncertain about the effect of Perinatal Mobile Apps on Social support at 1 month. |
| State anxiety [31] assessed with the STAI-Y1 Scale from 20 to 80 points (1 month) | 327 (1 RCT) | n = 176 Mean: 37.9 | n = 151 Mean: 39.4 | MD: -1.50 (-3.93, 0.93) | ⊕⊕◯◯ Low (a) | Perinatal mobile apps may result in little to no difference in state anxiety at 1 month. |
| Parent-to-infant Bonding Questionnaire [31] Scale from 0 to 21 points (1 month) | 327 (1 RCT) | n = 176 Mean: 3.14 | n = 151 Mean: 3.39 | MD: -0.25 (-0.92, 0.42) | ⊕⊕◯◯ Low (a) | Perinatal Mobile apps may result in little to no difference in parent-to-infant bonding at 1 month. |
| Breastfeeding Self-Efficacy scale [30] Scale from14 to 70 points (4–6 weeks) | 439 (1 RCT) | n = 224 Mean: 48.3 | n = 215 Mean: 47.4 | MD: 0.90 (-2.30, 4.10) | ⊕⊕◯◯ Low (a) | Perinatal mobile apps may result in little to no difference in breastfeeding self-efficacy at 4–6 weeks. |
| Partner support during breastfeeding [30] Scale from 25-100 points (4–6 weeks) | 439 (1 RCT) | n = 224 Mean: 82.8 | n = 215 Mean: 81.7 | MD: 1.10 (-2.48, 4.68) | ⊕⊕◯◯ Low (a) | Perinatal mobile apps may result in little to no difference in partner support during breastfeeding at 4–6 weeks. |

RCT: Randomized Controlled Trial; SMD: Standardized Mean Difference; MD: Mean Difference; CI: Confidence Interval; MID: Minimal Important Difference; STAI: State-Trait Anxiety Inventory.

*MIDs for SMDs*: 0.2 for postnatal depression, perceived self-efficacy, parenting satisfaction and social support. partner support during breastfeeding,

*MIDs for MDs*: 10 points for state anxiety, 2 points for parent-to-infant bonding, 5 points for breastfeeding self-efficacy and 7 points for Partner support during breastfeeding.

Explanations of certainty of evidence:

[a] Two levels of certainty were decreased for risk of bias.

[b] Two levels of certainty were decreased for imprecision

[c] One level of certainty was decreased for imprecision

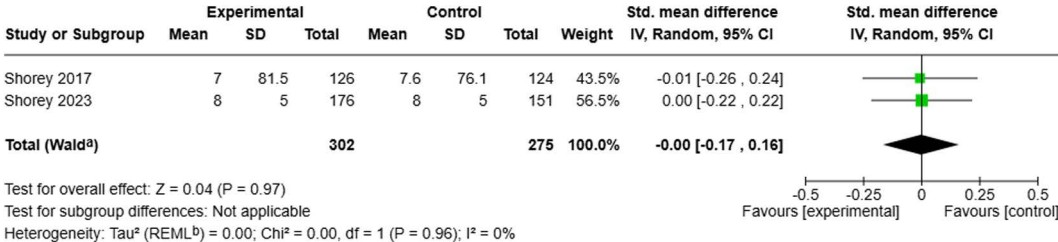

**Fig 3. Meta-analysis results: Effect of perinatal mobile applications (Apps) on postnatal depression.** Shorey (2017) reported percentage changes, while Shorey (2023) reported final outcomes. Both studies used the Edinburgh Postnatal Depression Scale (EPDS) and included results for both fathers and mothers at a one-month follow-up.

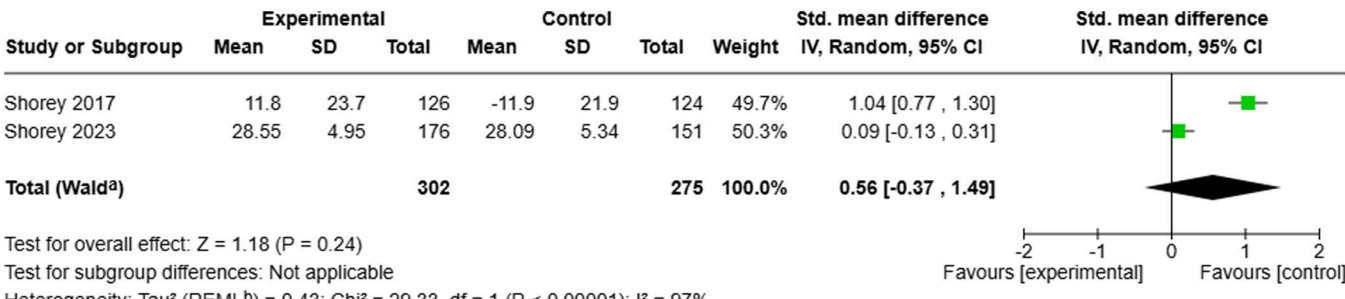

**Fig 4. Meta-analysis results: Effect of perinatal mobile applications (Apps) on perceived self-efficacy.** Shorey (2017) reported percentage changes, while Shorey (2023) reported final outcomes. Both studies used the Parenting Self-Efficacy (PSE) scale and included results for both fathers and mothers at a one-month follow-up.

their effects on perceived parenting self-efficacy, parenting satisfaction, and social support. The main limitations of the studies included inadequate blinding of participants, personnel, and outcome assessors, along with incomplete outcome data.

## Overall completeness and applicability of the evidence

When considering the extrapolation of our findings, several factors must be taken into account. The RCTs included in this review were conducted in three regions: one in America (United States) [28], two in Asia (Singapore) [29,31], and one in Oceania (Australia) [30]. Additionally, the studies included in this review were conducted in high-income and upper-middle-income countries, potentially limiting the generalizability of the findings to low-resource settings. Further research is needed to evaluate the effectiveness of perinatal mobile applications in diverse socioeconomic and cultural contexts.

Another important aspect is the participation of both mothers and fathers in the included studies. Some studies used the same assessment instruments for both parents, even when these tools were only validated for mothers and not for fathers. Additionally, some studies did not stratify results by parent gender, limiting the ability to evaluate potential differential effects [28,30].

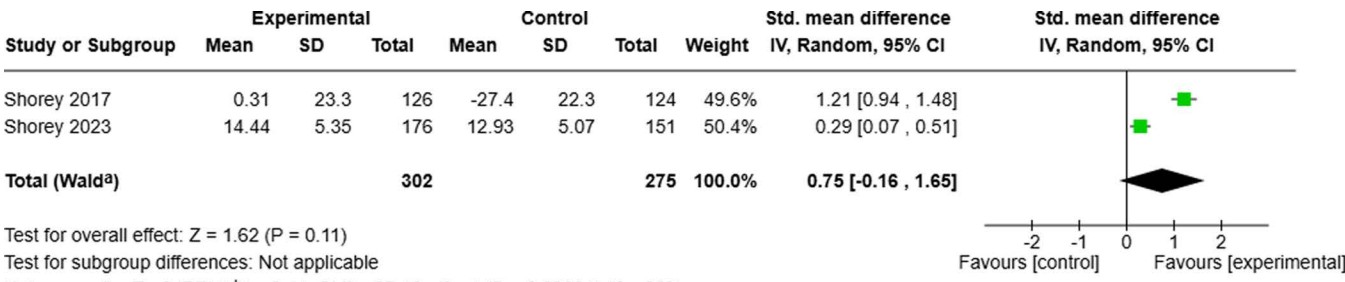

| Study or Subgroup | Experimental | | | Control | | | Weight | Std. mean difference IV, Random, 95% CI |
|---|---|---|---|---|---|---|---|---|
| | Mean | SD | Total | Mean | SD | Total | | |
| Garfield 2016 | 77.4 | 13.281566 | 30 | 70.6 | 7.776889 | 31 | 32.7% | 0.62 [0.10 , 1.13] |
| Shorey 2017 | 2.6 | 16.8 | 126 | -35.4 | 13.3 | 124 | 33.5% | 2.50 [2.17 , 2.83] |
| Shorey 2023 | 80.85 | 14.71 | 176 | 80.55 | 13.33 | 151 | 33.8% | 0.02 [-0.20 , 0.24] |
| Total (Wald[a]) | | | 332 | | | 306 | 100.0% | 1.05 [-0.43 , 2.52] |

Test for overall effect: Z = 1.39 (P = 0.16)
Test for subgroup differences: Not applicable
Heterogeneity: Tau² (REML[b]) = 1.66; Chi² = 149.95, df = 2 (P < 0.00001); I² = 98%

**Footnotes**
[a]CI calculated by Wald-type method.
[b]Tau² calculated by Restricted Maximum-Likelihood method.

**Fig 5. Meta-analysis results: Effect of perinatal mobile applications (Apps) on parenting satisfaction.** Shorey (2017) reported percentage differences between baseline and final scores, while Shorey (2023) reported final outcomes. Both used the "What Being the Parent of a New Baby is Like" (WPBL) scale. Garfield (2016) used the Parenting Sense of Competence Scale (PSOC) and reported final outcomes. All studies evaluated both mothers and fathers, with results reported at four weeks.

| Study or Subgroup | Experimental | | | Control | | | Weight | Std. mean difference IV, Random, 95% CI |
|---|---|---|---|---|---|---|---|---|
| | Mean | SD | Total | Mean | SD | Total | | |
| Shorey 2017 | 0.31 | 23.3 | 126 | -27.4 | 22.3 | 124 | 49.6% | 1.21 [0.94 , 1.48] |
| Shorey 2023 | 14.44 | 5.35 | 176 | 12.93 | 5.07 | 151 | 50.4% | 0.29 [0.07 , 0.51] |
| Total (Wald[a]) | | | 302 | | | 275 | 100.0% | 0.75 [-0.16 , 1.65] |

Test for overall effect: Z = 1.62 (P = 0.11)
Test for subgroup differences: Not applicable
Heterogeneity: Tau² (REML[b]) = 0.41; Chi² = 27.10, df = 1 (P < 0.00001); I² = 96%

**Footnotes**
[a]CI calculated by Wald-type method.
[b]Tau² calculated by Restricted Maximum-Likelihood method.

**Fig 6. Meta-analysis results: Effect of perinatal mobile applications (Apps) on social support.** Shorey (2017) assessed partner support and reported percentage differences in scores, while Shorey (2023) assessed both partner support and support from others, reporting final scores. Both studies surveyed mothers using the Perceived Social Support for Parenting scale, with a one-month follow-up.

Regarding the interventions, all studies evaluated mobile applications focused on psychoeducational support, providing structured information and content to parents of newborns without synchronous bidirectional interaction. However, these interventions differ in application functionality: NICU-2-Home [28] included parental mood tracking and guidance on infant care. Supportive Parenting App [31] implemented a peer support model, pairing participants with trained volunteers to offer emotional and informational support throughout the intervention. Meanwhile, Milk Man [30] provided discussion forums where users could interact with each other. Finally, Home-but not alone [29] included asynchronous communication with healthcare professionals until participants gained access to the application. This heterogeneity may affect the comparability of results and limit the generalizability of conclusions.

The timing of outcome measurement varied across studies, ranging from immediately after the intervention to six weeks postpartum, limiting the ability to assess long-term effects. The absence of extended follow-up further hinders the evaluation of sustained benefits, including potential late-emerging effects or intervention decay over time. This limitation

also restricts insights into whether the observed outcomes translate into meaningful, lasting improvements in clinical or behavioral measures.

### Certainty of the evidence

The certainty of the evidence ranged from low to very low, primarily due to risk of bias and imprecision.

Regarding the risk of bias, most of the studies included in this review exhibited concerns across multiple domains, particularly in participant and personnel blinding, as well as outcome assessment. A high risk of bias was identified in blinding of participants or blinding of personnel across several studies [29,30], which may compromise the validity of the findings. Since blinding in digital interventions poses methodological challenges, future research may explore strategies such as the use of sham applications or active comparator interventions to minimize placebo effects and prevent systematic differences in group behavior [19,41].

Additionally, concerns were identified in outcome assessment and the management of incomplete data, potentially introducing further biases in the interpretation of intervention effects. However, random sequence generation and mitigation of selective reporting bias were deemed adequate in most studies, partially reducing the overall risk of bias. Overall, while some studies exhibited a low risk of bias in certain areas, key methodological limitations reinforce the need for cautious interpretation of the findings.

### Agreements and disagreements with other studies or reviews

Our systematic review evaluated the effects of perinatal mobile applications on psychosocial and parenting outcomes in parents of newborns without gestational age restrictions, encompassing a diverse range of contexts and populations.

Three previous systematic reviews have examined the effects of digital interventions during the perinatal period (Feng et al. [13]; Kermani et al. [14]; Chua et al. [5]). However, their assessment of interventions targeting couples (both mothers and fathers) was limited, as each included only one or two RCTs. All three reviews incorporated Garfield et al. [28], while one also included 2017 Shorey et al. [29]. However, none of these reviews considered Scott et al. [30] or 2023 Shorey et al. [31], all of which were included in our review.

Two of these reviews (Kermani et al. [14]; Chua et al. [5]) included studies involving mothers or couples. While Feng et al. [13] focused on couples, it also included a study where apps were used primarily for contacting medical personnel, which falls outside the scope of our review. Additionally, all three reviews included non-randomized studies, which are more susceptible to bias, limiting the reliability of their findings and reducing the strength of causal inferences. Of note, any of these reviews perform meta-analysis or assess the certainty of the evidence.

This may explain why these reviews reported positive effects of app-based interventions on partner support during breastfeeding [5], reduction of depressive symptoms [13], and enhanced parental confidence in newborn care [14], whereas our study did not find relevant improvements in these outcomes.

### Limitations and strengths

This systematic review has some limitations. Despite a comprehensive search across multiple databases, some relevant studies may have been missed due to lack of indexing or limited accessibility. The limited number of randomized clinical trials restricts the generalizability of the findings; and the heterogeneity in the approaches, functionalities, and levels of interaction of the applications complicates result comparisons.

The inconsistency in outcome measurement represents an additional limitation, as different instruments were used across studies, making direct comparisons difficult. Furthermore, variability in how outcomes are reported—some studies focusing on parents or mothers individually and others on couples—pose challenges for comparison and meta-analysis. The limited variability among the studies prevented a thorough analysis of heterogeneity and sensitivity analysis. This restricts the ability to explore the causes of heterogeneity and assess the robustness of the results.

However, this review has significant strengths. To the best of our knowledge, it is the first systematic review with a meta-analysis to evaluate the effects of perinatal mobile applications on psychosocial and parenting outcomes in parents, without gestational age restrictions. A comprehensive search was conducted without language restrictions, enhancing the representativeness of the existing literature. However, we did not find any studies published in languages other than English. Study selection and data extraction were performed in duplicate to minimize errors and biases. Furthermore, this review is among the few that apply the GRADE methodology, providing a structured and transparent assessment of the certainty of the evidence.

## Conclusion

In this systematic review of randomized controlled trials, the evidence suggests that perinatal mobile applications may have little to no effect on postnatal depression, state anxiety, parent-to-infant bonding, breastfeeding self-efficacy, and partner support during breastfeeding. Furthermore, the evidence is very uncertain regarding their effects on perceived parenting self-efficacy, parenting satisfaction, and social support. Rigorous studies with long-term follow-up are needed to determine the actual impact of these interventions on psychosocial and parenting outcomes.

### Declaration of generative AI and AI-assisted technologies in the writing process

During the preparation of this manuscript, the authors used ChatGPT 4.0 to assess the readability of the draft and provide feedback, which was considered by the authors. After using this tool, authors reviewed and edited the content as needed, assuming full responsibility for the final content of the publication.

## Supporting information

**S1 Appendix. PRISMA 2020 checklist.**
(DOCX)

**S2 Appendix. R script for extracting data from published figures.**
(DOCX)

**S1 File. Protocol. PROSPERO registration.**
(DOCX)

**S1 Table. Search strategy.**
(DOCX)

**S2 Table. Excluded studies reviewed at full text stage.**
(DOCX)

**S3 Table. Data extraction sheet for included studies.**
(XLSX)

## Author contributions

**Conceptualization:** Luis Ttito-Paricahua, Marlene Magallanes-Corimanya, Liz Mendoza-Aucaruri, Jean Pierre López-Mesia, Evelyn M. Asencios-Falcón, Alicia Lopez-Gomero, Alvaro Taype-Rondan.

**Data curation:** Luis Ttito-Paricahua, Marlene Magallanes-Corimanya, Liz Mendoza-Aucaruri, Jean Pierre López-Mesia, Evelyn M. Asencios-Falcón, Alicia Lopez-Gomero, Alvaro Taype-Rondan.

**Formal analysis:** Luis Ttito-Paricahua, Marlene Magallanes-Corimanya, Liz Mendoza-Aucaruri, Jean Pierre López-Mesia, Alvaro Taype-Rondan.

**Investigation:** Luis Ttito-Paricahua, Marlene Magallanes-Corimanya, Liz Mendoza-Aucaruri, Jean Pierre López-Mesia, Evelyn M. Asencios-Falcón, Alicia Lopez-Gomero, Alvaro Taype-Rondan.

**Methodology:** Luis Ttito-Paricahua, Marlene Magallanes-Corimanya, Liz Mendoza-Aucaruri, Jean Pierre López-Mesia, Evelyn M. Asencios-Falcón, Alicia Lopez-Gomero, Alvaro Taype-Rondan.

**Project administration:** Luis Ttito-Paricahua, Marlene Magallanes-Corimanya, Liz Mendoza-Aucaruri, Jean Pierre López-Mesia, Evelyn M. Asencios-Falcón, Alicia Lopez-Gomero, Alvaro Taype-Rondan.

**Supervision:** Luis Ttito-Paricahua, Marlene Magallanes-Corimanya, Liz Mendoza-Aucaruri, Jean Pierre López-Mesia, Alvaro Taype-Rondan.

**Validation:** Alvaro Taype-Rondan.

**Visualization:** Luis Ttito-Paricahua, Marlene Magallanes-Corimanya, Liz Mendoza-Aucaruri, Jean Pierre López-Mesia, Evelyn M. Asencios-Falcón, Alicia Lopez-Gomero, Alvaro Taype-Rondan.

**Writing – original draft:** Luis Ttito-Paricahua, Marlene Magallanes-Corimanya, Liz Mendoza-Aucaruri, Jean Pierre López-Mesia, Evelyn M. Asencios-Falcón, Alicia Lopez-Gomero, Alvaro Taype-Rondan.

**Writing – review & editing:** Luis Ttito-Paricahua, Marlene Magallanes-Corimanya, Liz Mendoza-Aucaruri, Jean Pierre López-Mesia, Evelyn M. Asencios-Falcón, Alicia Lopez-Gomero, Alvaro Taype-Rondan.

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
