## [Decision Letter · Decision Letter 0]

23 Jul 2025

PMEN-D-25-00177

Effects of perinatal mobile apps for couples on psychosocial and parenting outcomes: A systematic review and meta-analysis

PLOS Mental Health

Dear Dr. Taype-Rondan,

Thank you for submitting your manuscript to PLOS Mental Health and I am sorry for the delay in reaching a decision - thank you for your patience. After careful consideration of the reviewer reports, we feel that your paper has merit but does not yet fully meet PLOS Mental Health’s publication criteria as it currently stands. Therefore, we invite you to submit a revised version of the manuscript that addresses the points raised during the review process.

Please address all of the points raised by the reviewers, which you can find below.

We look forward to receiving your revised manuscript.

Kind regards,

Karli Montague-Cardoso

Executive Editor

PLOS Mental Health

Journal Requirements:

i. Please clarify all sources of funding (financial or material support) for your study. List the grants (with grant number) or organizations (with url) that supported your study, including funding received from your institution. 

ii. State the initials, alongside each funding source, of each author to receive each grant.

iii. State what role the funders took in the study. If the funders had no role in your study, please state: “The funders had no role in study design, data collection and analysis, decision to publish, or preparation of the manuscript.”

iv. If any authors received a salary from any of your funders, please state which authors and which funders.

2. We noticed that you used “unpublished” in the manuscript. We do not allow these references, as the PLOS data access policy requires that all data be either published with the manuscript or made available in a publicly accessible database. Please amend the supplementary material to include the referenced data or remove the references.

3. As required by our policy on Data Availability, please ensure your manuscript or supplementary information includes the following: 

4. In the online submission form, you indicated that “The underlying data for this study are available in the supplementary information of the manuscript. Interested parties can request the data directly from the corresponding author via email. Additionally, the protocol for this study is registered in PROSPERO with the identifier CRD42024578397, accessible at the following link: (https://www.crd.york.ac.uk/PROSPERO/)”. 

3. Uploaded as supplementary information.

5. Some material included in your submission may be copyrighted. According to PLOS’s copyright policy, authors who use figures or other material (e.g., graphics, clipart, maps) from another author or copyright holder must demonstrate or obtain permission to publish this material under the Creative Commons Attribution 4.0 International (CC BY 4.0) License used by PLOS journals. Please closely review the details of PLOS’s copyright requirements here: PLOS Licenses and Copyright. If you need to request permissions from a copyright holder, you may use PLOS's Copyright Content Permission form.

Potential Copyright Issues:

We do not publish any copyright or trademark symbols that usually accompany proprietary names, eg (R), (C), or TM (e.g. next to drug or reagent names). Therefore please remove all instances of trademark/copyright symbols throughout the text, including Rayyan® on page 5.

Additional Editor Comments (if provided):

Reviewers' comments:

Reviewer's Responses to Questions

**Comments to the Author**

1. Does this manuscript meet PLOS Mental Health’s publication criteria ? Is the manuscript technically sound, and do the data support the conclusions? The manuscript must describe methodologically and ethically rigorous research with conclusions that are appropriately drawn based on the data presented.

Reviewer #1: Yes

Reviewer #2: Yes

2. Has the statistical analysis been performed appropriately and rigorously?

Reviewer #1: Yes

Reviewer #2: Yes

3. Have the authors made all data underlying the findings in their manuscript fully available (please refer to the Data Availability Statement at the start of the manuscript PDF file)?

Reviewer #1: Yes

Reviewer #2: Yes

4. Is the manuscript presented in an intelligible fashion and written in standard English?

Reviewer #1: Yes

Reviewer #2: Yes

5. Review Comments to the Author

Reviewer #1: Introduction is well-structured!

Methods

- Revise the full-form of PROSPERO (International prospective register of systematic reviews)

- State research question in 'Research question and eligibility criteria' section.

- Even though you have a separate supplementary file for search strategies utilized for each databases, briefly describe major search terms in the 'search strategy' section.

- Can you give a bit more information about standardized data extraction form? Was there any specific form or did you make form in Excel?

- Risk of bias assessment: Cochrane risk-of-bias tool for randomized trials (RoB 2) is the recommended tool to assess the risk of bias in RCTs. Why did you use RoB 1.0?

- Certainty of the evidence assessment: State full form of STAI scale. Correct spelling of 'literature' before citation [26].

Results

- While there is a separate figure 1 for study selection. I would suggest authors to include a brief description of selection (e.g., how many duplicates removed, how many were screened for full-text, etc.).

- I believe this review is focused on effect of mobile app (app based intervention) but fourth study (Abbass-Dick 2019) is related to providing intervention through website (eHealth Breastfeeding Website). Can you please explain how did this study meet your inclusion criteria?

- In addition to describing what study used what scales (e.g., EDPS, PPSS), I suggest authors to provide appropriate reference for those scales.

- In table 2, in the first column (outcomes), suggest citing those studies which reported each outcomes (e.g., cite 2 RCTs reporting postnatal depression).

- Include results of meta-analyses below each figure headings.

Strengths and Limitations

- In strengths, authors mentioned that search was conducted without

language restrictions. Did this review found studies that were conducted in language other than English? If not, it may not be suitable to include that as an strength.

Reviewer #2: This article is remarkably precise and well-composed. I commend the authors for their meticulous attention to detail and clarity throughout the manuscript.

I have one suggestion to further enhance the article: in certain paragraphs—such as the one cited below—it would be beneficial to include individual references for each key component or claim. Citing a single reference at the end of a paragraph may lead to ambiguity regarding the source of specific information. Providing direct citations for each element will strengthen the transparency and traceability of the evidence presented.

"we evaluated five key domains: risk of

bias (using the Cochrane risk of bias tool), imprecision (considering whether

confidence intervals included one or two MIDs), indirectness (by comparing our

PICO question (population, intervention, comparison, outcome) with those of the

included studies), inconsistency (assessing whether studies point results were

distributed on one or both sides of an MID), and publication bias (examined in

meta-analyses with at least 10 studies) [23]."

6. PLOS authors have the option to publish the peer review history of their article (what does this mean? ). If published, this will include your full peer review and any attached files.

**Do you want your identity to be public for this peer review?** For information about this choice, including consent withdrawal, please see our Privacy Policy .

Reviewer #1: No

Reviewer #2: **Yes: ** Azam Aslani

---

## [Editor Report · Decision Letter 1]

21 Aug 2025

Effects of perinatal mobile apps for couples on psychosocial and parenting outcomes: A systematic review and meta-analysis

PMEN-D-25-00177R1

Dear Dr Taype-Rondan,

We are pleased to inform you that your manuscript 'Effects of perinatal mobile apps for couples on psychosocial and parenting outcomes: A systematic review and meta-analysis' has been provisionally accepted for publication in PLOS Mental Health.

Best regards,

Karli Montague-Cardoso

Staff Editor

PLOS Mental Health